# EMERGENCE OF MACHINE LANGUAGE IN LLM-BASED AGENT COMMUNICATION

## ABSTRACT

Language emergence is a hallmark of human intelligence, as well as a key indicator for assessing artificial intelligence. Unlike prior studies grounded in multi-agent reinforcement learning, this paper asks whether machine language, potentially not human-interpretable, can emerge between large language model (LLM) agents. We study this in the stylish paradigm of referential games, where a speaker describes a target object into a message with a predefined alphabet, and a listener, given the message, must identify the target among distractors. We propose an agent design that enables the speaker to retrieve semantically similar words before composing a message, and the listener to decode the message based on structural proximity between words. We observe that even given a set of $541$ objects, the two agents successfully develop a shared language: they acquire meanings for each object through only 4 rounds of communication, with at most 3 attempts per communication. Additionally, analyses reveal that the emergent language exhibits compositionality, generalizability, morphemes, and polysemy, which are defining features of human language. Our project can be accessed via the following link: https://anonymous.4open.science/r/ELofLLM-1746/.

## 1 INTRODUCTION

Language is a hallmark of human intelligence (Peters et al., 2025; Lazaridou & Baroni, 2020). Its development relies on several advanced cognitive and social abilities, including *social learning* (Snow, 2013), where individuals associate meanings with arbitrary symbols through shared understanding, *syntactic reasoning* (Ramer, 1976), which allows us to structure words to convey meaning, and *mental flexibility* (Jacques & Zelazo, 2005), allowing the creation of infinite expressions from a finite set of elements. Given these complexities, it is no surprise that enabling the emergence of language has been a significant challenge for AI over the years, and continues to serve as a crucial yardstick for assessing progress toward artificial general intelligence (Peters et al., 2025; Cowen-Rivers & Naradowsky, 2020).

Recent studies on language emergence in AI primarily focus on multi-agent reinforcement learning (MARL) (Foerster et al., 2016; Havrylov & Titov, 2017). A number of approaches have been proposed to enable agents to develop language that supports not only agent-to-agent but also agent-to-human communication, mirroring the characteristics of human natural language (Wagner et al., 2003; Bouchacourt & Baroni, 2018; Noukhovitch et al., 2021). Notable studies such as DIAL (Foerster et al., 2016), CommNet (Sukhbaatar et al., 2016), and IC3Net (Singh et al.) enabled agents to acquire differentiable communication in an end-to-end manner, driven by the objective of maximizing task rewards.

In this paper, we explore whether language can emerge through communication between large language model (LLM)-based agents. Unlike MARL agents, LLM-based agents are inherently able to understand and generate human natural language, due to their extensive pre-training on vast human corpora (Gao et al., 2024; Ren et al., 2024; Wang et al., 2024). Thus, while communication between these agents using human natural language might seem trivial, the true challenge—and the focus of this study—lies in investigating the emergence of *machine language* that does *not* exist in training data *nor* is interpretable by humans. Specifically, we ask: Are LLM-based agents able to develop machine language through interactions? If so, does the emergent machine language exhibit

characteristics typical of human natural language, which are defining features of human cognitive abilities?

We explore these questions within the framework of referential games (Lewis, 1969), a well-established paradigm for studying language emergence (Figure 1). In this game, two agents are randomly assigned roles as either a speaker or a listener. Given an arbitrary object defined by a set of semantic features, the speaker creates a symbolic word for the object using letters from a pre-defined alphabet, deliberately avoiding human natural language. The listener, upon receiving the word, attempts to identify the object it corresponds to, selecting it from a set of distractors. Machine language emerges if the two agents can successfully communicate—measured by the listener's accurately identifying the object the speaker intends to convey—across multiple objects. For agent behaviors, we make simple yet principled assumptions: agents are allowed to retrieve and utilize words from their vocabulary that share similar semantic features or structural proximity. This design aims to keep the system minimal, avoiding unnecessary complexity, and allowing us to focus on understanding, at the simplest level, whether and how LLM-based agents can develop machine language.

In our experiments, we consider a total of 541 objects and two variants of alphabets, where each letter is randomly represented by 2-3 characters. Machine language is considered to emerge if the two agents achieve successful communication on the majority of 400 objects, using the alphabet that is not interpretable by humans. We find that a machine language successfully emerges within the referential game, starting from successful communication of 268 objects (67%) by the first round, to 388 objects (97%) by the third round, and all 400 objects (100%) by the fourth round. Within only four rounds, agents achieve successful communication on all objects, highlighting the efficiency of the emergent language. Additionally, to evaluate whether the emergent language enables agents to communicate about objects never seen before, we introduced 141 novel objects beyond the previous four rounds, and found that agents could immediately communicate about most of them using the emergent language. Further analysis shows that the emergent language also exhibits hallmarks of human natural language, such as *compositionality* (e.g., the ability to create "apple-pie" based on the meanings of "apple" and "pie"), *morphemes*, and *polysemy*. Finally, we examine key factors that influence language emergence, including alphabet size, maximum word length, and number of objects, and show how these factors affect the quality of emergent machine language. The repository is accessible via the following link: `https://anonymous.4open.science/r/ELofLLM-1746/`.

In summary, our key contributions are as follows:

1. We present a novel paradigm to investigate language emergence between LLM-based agents, focusing on their ability to develop language while minimizing the influence of human semantic knowledge embedded in training data.

2. We present an agent design with minimal complexity that allows agents to retrieve and utilize words from their vocabulary based on semantic similarity and structural proximity, ultimately leading to emergence of machine languages.

3. We show that the emergence of machine language is efficient and robust. Moreover, the emergent language enjoys generalizability, compositionality, morphemes and polysemy, which are features typical of human natural language and their cognitive abilities.

## 2 RELATED WORK

Communication between AI systems has been a long-standing challenge . Earlier studies have examined agents' symbol-based communication using carefully designed experimental simulations with simple, largely hand-crafted agents (Steels, 1997; Nowak & Krakauer, 1999; Cangelosi & Parisi, 2002; Christiansen & Kirby, 2003). For example, Batali (1998) conducted simulations in which a speaker agent, given a binary vector representing the meaning of a simple phrase, encodes it as a sequence of characters. In recent years, MARL is also employed to explore how language can emerge between RL-based agents in partially observable environments (Lazaridou et al., 2016; Foerster et al., 2016; Lee et al., 2017; Kajić et al., 2020). Foerster et al. (2016) introduces two deep learning approaches that enable agents to learn communication protocols in end-to-end manner using centralized learning with decentralized execution. Havrylov & Titov (2017) investigated the emergence of natural language in referential games, where agents develop discrete, symbol-based

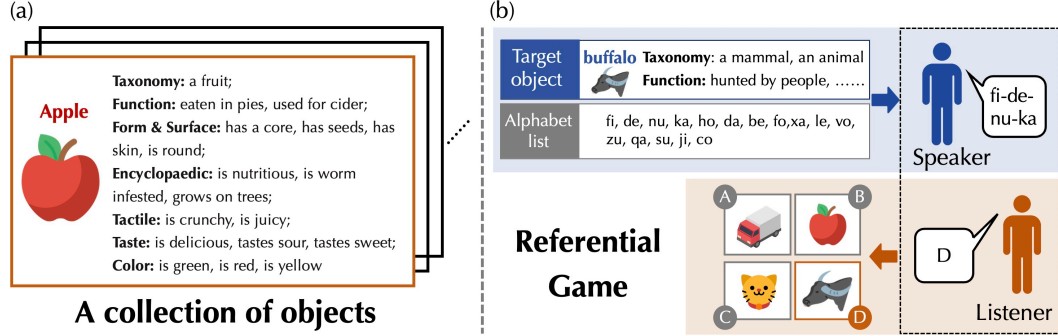

Figure 1: Schematic illustration of object representation and referential games. (a) An apple is used to illustrate how objects are represented as a set of semantic features rather than by their lexical label in natural language. (b) In the referential game paradigm, the speaker encodes a target object, represented through semantic features, into a symbolic message using a given alphabet, and the listener must identify the target object from among multiple distractors.

communication protocols. Kajić et al. (2020) showed that agents in cooperative gridworld navigation tasks can develop interpretable communication protocols through spatially grounded signals and basic compositional structures, improving coordination efficiency. Li et al. (2024) proposed a language-grounded communication framework, enabling zero-shot generalization in ad-hoc teamwork scenarios. Unlike most prior studies grounded in hand-crafted simulations or MARL, our work explores language emergence in LLM-based agents and introduces a new paradigm of machine language emergence, which allows for focusing on agents' ability to develop language while minimizing the influence of human semantic knowledge embedded in LLMs' training data.

It is worth mentioning that, more recently, a few studies examined language evolution in the context of LLM-based agents. Kouwenhoven et al. (2025) explored how inductive biases in LLMs help maintain alignment with humans, thereby shaping a shared language for successful human–machine interaction. Ashery et al. (2025) investigated whether LLM-based agents could spontaneously bootstrap social conventions in a naming game, establishing consistent references to individuals' names from a given naming pool. In contrast to our work, which explores language emergence from scratch, these studies assume a pre-defined vocabulary for each agent, where language already exists from the outset. To the best of our knowledge, our work is the first to demonstrate that machine language—non-interpretable by humans—can emerge from scratch between LLM-based agents, while also exhibiting defining features of human natural language.

## 3 PRELIMINARY

**Object representation.** Language is a system of communication in which words organized by grammatical structure to transmit information (Simpson & Weiner, 1989). Such information, at its simplest, can be an *object*—for instance, an apple, a car, or any other tangible entity. Since agents may only perceive objects in context, the *brain region taxonomy* offers a way to conceptualize them beyond their lexical labels (e.g., the word "apple"). It represents objects through ten categories of semantic features (Cree & McRae, 2003): *color, form and surface, motion, smell, sound, tactile, taste, function, encyclopedic, and taxonomy*. Hence, an object is represented as a combination of characteristic features rather than by its lexical label. For example, as shown in Figure 1(a), the object "apple" can be represented *color: is red / green*, *taste: tastes sweet*, *tactile: is juicy*, etc. Such representations capture the perceptual and functional essence of the object without naming it explicitly.

**Referential Game.** The *referential game* is a widely used paradigm for studying language emergence, involving two agents that interact with one another (Figure 1(b)). At the start of each round, the agents are randomly assigned the roles of *speaker* and *listener*. The speaker encodes a target

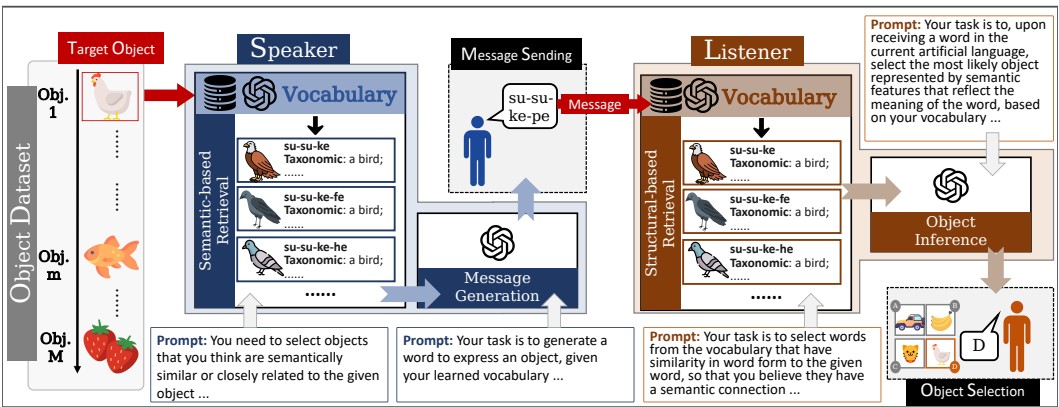

Figure 2: LLM-based agent communication. Given a target object defined by a set of semantic features, the speaker retrieves words from memory that share similar semantic features and generates a symbolic message using a predefined alphabet. The listener, upon receiving the message, retrieves words from memory based on structural proximity and attempts to infer the target object by selecting it from a set of distractors. Communication is considered successful if the listener correctly identifies the target object. A language emerges when communication about most objects is consistently successful.

object—for example, a buffalo—into a symbolic message (e.g. "fi-de-nu-ka") using an alphabet $\mathcal{A}$. After receiving this message, the listener must identify the target object among several distractors.

Formally, let $\mathcal{O}$ denote a set of objects, with each generic object $o$ defined by a set of $n$ semantic features $f_o = \{f_o^i\}_{i=1}^n$. The speaker is assigned a target object $o_t \in \mathcal{O}$ and produces a symbolic word $w_t$ from the alphabet $\mathcal{A}$. Then, given some distractors $o' \in \mathcal{O}, o' \neq o_t$, the listener aims to distinguish $o_t$ through the word $w_t$ sent by the speaker. A communication is successful if the listener can correctly identify the target object. A language is considered to emerge if communications about most objects in the set $\mathcal{O}$ are successful (Lazaridou et al., 2018). In general, the emergence of new natural-language-like languages from referential games without introducing a priori design is non-trivial (Rita et al., 2022; Steinert-Threlkeld, 2020).

**Characteristics of language.** To evaluate the quality of an emergent language among individuals, researchers typically focus on two key characteristics: (i) *Compositionality*, which refers to the principle that the meaning of a complex expression is determined by the meanings of its parts and the manner of their combination (Partee et al., 1995). For example, we can infer that "apple pie" is a pie made from apples because the expression is composed of "apple" and "pie"; (ii) *Generalizability*, which is the ability to extend learned linguistic structures to novel combinations, thereby supporting hierarchical descriptions of concepts and relations (Chaabouni et al., 2021; Mu & Goodman, 2021). For instance, if senders have learned to describe a "red square" and a "blue circle", they should also be able to generalize and understand a "red circle" or a "blue square" without explicit training.

## 4 LLM-BASED AGENT COMMUNICATION

In this section, we present an agent design for the LLM-based agent in referential games. We first describe how the agent's memory is utilized and updated, and then introduce the design from two perspectives: the speaker and the listener. An overview of our agent design is shown in Figure 2. Detailed prompts used for the LLMs are provided in Appendix A.

### 4.1 MEMORY

Memory is fundamental to language learning; without it, language cannot be learned or used consistently (Corballis, 2019; Ullman, 2004). Each LLM agent is equipped with a memory $M$ that stores a

vocabulary of $m$ object–word pairs $(o_j, w_j)_{j=1}^m$. Initially, both agents' memories are empty. As successful communications accumulate, memory is updated by storing new object–word pairs. When a new communication on an object $o_t$ begins, the speaker first checks whether $o_t$ already has an associated symbolic word $w_t$ in memory, which is searched through exact matching of its semantic features $f_t$. For the listener, upon receiving a symbolic word $w_t$, the most direct strategy is to search memory by string matching. When a match is found, the listener successfully identifies the target object among distractors, and both agents update their memories. Otherwise, the speaker generates a new message using semantically similar words, and the listener attempts to decode it based on structural proximity between words. These will be described in detail in Section 4.2 and 4.3.

## 4.2 SPEAKER: MESSAGE GENERATION WITH SEMANTIC SIMILARITY

When no symbolic word in the vocabulary encodes the target object $o_t$, we make the simple yet principled assumption that the speaker retrieve and utilize words from their vocabulary that share similar semantic features with $o_t$, a process inspired by how humans create new words (Dell, 1986; Levelt, 1993). To achieve this, we first instruct the speaker to retrieve similar object–word pairs based on the target object's semantic features, and then prompt it to generate a symbolic message for $o_t$ based on the retrieved pairs.

**Semantic-based Retrieval.** To minimize LLMs' reliance on inherent linguistic knowledge, objects are represented by semantic features rather than lexical labels in natural language. This enables the speaker to retrieve semantically similar object–word pairs by comparing the semantic features of the target object $o_t$ with those of the $m$ pairs $(o_j, w_j)_{j=1}^m$ stored in memory $M$. Specifically, we prompt LLMs to identify object–word pairs by checking for similarity in certain features, such as when the "taxonomy" of both objects is "a fruit". The speaker then retrieves and outputs a set of words that share similar semantic features with $o_t$ for message generation. This process can be represented as an LLM-based operation: $(o_s, w_s)_{s=1}^c \leftarrow \texttt{SemanticRetrieval}\big(o_t, (o_j, w_j)_{j=1}^m\big)$ where $(o_s, w_s)_{s=1}^c$ denotes the $c$ pairs retrieved from memory $M$.

**Message Generation.** Given the retrieved semantically similar pairs $(o_s, w_s)_{s=1}^c$, the speaker generates a symbolic message from a predefined alphabet $\mathcal{A}$ to describe the target object $o_t$. We prompt the LLM to reference the retrieved pairs when composing the message and to output a symbolic word $w_t$ with a maximum length of $\mathcal{L}$. Formally, this process is defined as an LLM-based operation: $w_t \leftarrow \texttt{MessageGeneration}\big(o_t, (o_s, w_s)_{s=1}^c\big)$. In cases where no semantically similar words are available (e.g., when the memory is empty initially), the speaker instead produces a symbolic word by randomly sampling from $\mathcal{A}$, with its length constrained by $\mathcal{L}$.

It is non-trivial to communicate using a novel pre-defined alphabet without any prior linguistic grounding, as agents must autonomously develop a machine language from scratch. Consequently, communication failures are common, particularly at the beginning when only few object–word pairs are available for reference. When a communication attempt fails, we prompt the speaker to learn from the failure by regenerating a message not only based on the words retrieved through similar semantic features, but also by referencing previously failed symbolic messages. Each agent is allowed at most $\mathcal{T}$ attempts to communicate. If all $\mathcal{T}$ attempts fail, the object is skipped for communication.

## 4.3 LISTENER: OBJECT SELECTION WITH STRUCTURAL PROXIMITY

When encountering an unfamiliar word $w_t$ sent by the speaker, we assume a simple principle: the listener infers its meaning based on the structural proximity between $w_t$ and the words in the vocabulary, and then attempts to distinguish the target object $o_t$ from multiple distractors. To this end, we first instruct the listener to retrieve similar object–word pairs based on the structure of $w_t$, and then we prompt the listener to select the target object from the distractors. Communication is considered successful once the listener identifies $o_t$ based on the message $w_t$ sent by the speaker.

**Structure-based Retrieval.** For the listener, the only way to infer the meaning of an unfamiliar message is to compare structural proximity between the word $w_t$ and those $m$ object-word pairs $(o_j, w_j)_{j=1}^m$ stored in memory $M$. Specifically, we prompt the LLM to retrieve structurally sim-

ilar object–word pairs based on the form of $w_t$. For example, "su-ke-he" and "su-ke-he-fe" are structurally proximate words. The listener then retrieves and outputs a set of structurally similar pairs for object inference. Formally, this process is represented as an LLM-based operation: $(o_p, w_p)_{p=1}^d \leftarrow \texttt{StructuralRetrieval}\big(w_t, (o_j, w_j)_{j=1}^m\big)$, where $(o_p, w_p)_{p=1}^d$ denotes the $d$ pairs retrieved from memory $M$.

**Object Inference.** Given the retrieved structurally similar pairs $(o_p, w_p)_{p=1}^d$, the listener attempts to infer the meaning of the symbolic message $w_t$, i.e., to infer the semantic features of the object symbolized by $w_t$. We prompt the LLM to reference the retrieved pairs when distinguishing among options and to select the object it infers to be the target $o_t$. Formally, this process is defined as an LLM-based operation:$o_l \leftarrow \texttt{ObjectInference}\big(w_t, (o_p, w_p)_{p=1}^d\big)$, where $o_l$ is the object selected by the listener. If $o_l = o_t$, communication is considered successful, as the listener has identified the correct object. In cases where no structurally similar words are available (e.g., when the memory is initially empty), the listener instead infers the object by randomly selecting from all available options.

## 5 EXPERIMENT

Our experimental study aims to address three key questions: (i) Are LLM-based agents able to develop machine language through interactions? (ii) If so, does the emergent machine language exhibit characteristics typical of human natural language, which are defining features of human cognitive abilities? (iii) What factors may influence its emergence? We outline the experimental setup in Section 5.1. We answer the following questions in Section 5.2. Experimental code is available at: `https://anonymous.4open.science/r/ELofLLM-1746/`.

### 5.1 SETTINGS

**Referential game.** Adapted from previous work (Havrylov & Titov, 2017; Van Eecke et al., 2022), our experiments have two stages: (i) *Stage 1: Repeated communication.* Agents play a repeated four-round referential game. One round covers all objects. For each object, agents are randomly assigned as either speaker or listener and have at most 3 attempts to communicate. In each attempt, the speaker encodes the target object (e.g., a buffalo) into a symbolic message (e.g., "fi-de-nu-ka") using a predefined alphabet. The listener must identify the target object among 4 distractors. If communication succeeds, they move to the next object. If all three attempts fail, the object is skipped. (ii) *Stage 2: Generalizability.* Agents play a one-round referential game with previously unseen objects. Since this stage tests whether the emergent language generalizes to new objects, words generated for unseen objects are not stored in agent memory. All experiments use `gpt-4.1-mini`. Details on model parameters and prompts are in Appendix A.

**Object dataset.** Following the brain region taxonomy, McRae et al. (2005) introduced a dataset of 541 objects, classified into 25 categories, including living types (e.g., fruits, mammals) and nonliving types (e.g., furniture, tools). Each object is described through ten categories of semantic features, without explicitly referring to its name in natural language. For example, as aforementioned in Figure 1(a), the object "apple" can be represented as *color: is red/green*, *taste: tastes sweet*, *tactile: is juicy*, etc. We randomly split the dataset into two subsets corresponding to the two game stages: (i) 400 objects in Stage 1, used to investigate whether agents can autonomously develop a machine language for communication, and (ii) 141 objects in Stage 2, used to evaluate whether the emergent language generalizes to previously unseen objects. Further details of the object representations are provided in Appendix B.

**Alphabet design.** We design an alphabet distinct from natural language to ensure that LLM-based agents communicate solely through symbols rather than relying on linguistic knowledge acquired from large human corpora (Devlin et al., 2019; Brown et al., 2020). We define an alphabet $\mathcal{A}$ consisting of 16 letters, with each letter being a *syllable* formed by a combination of *consonants* and *vowels* (Kirby et al., 2008). Specifically, we consider two types of letters: (i) consonant-vowel (CV) pairs (e.g., "ho", "da", "xa"), and (ii) vowel-consonant-vowel (VCV) triplets (e.g., "eno", "uza", "eca"). Words generated by agents contain at most six letters ($\mathcal{L} = 6$). For example, the

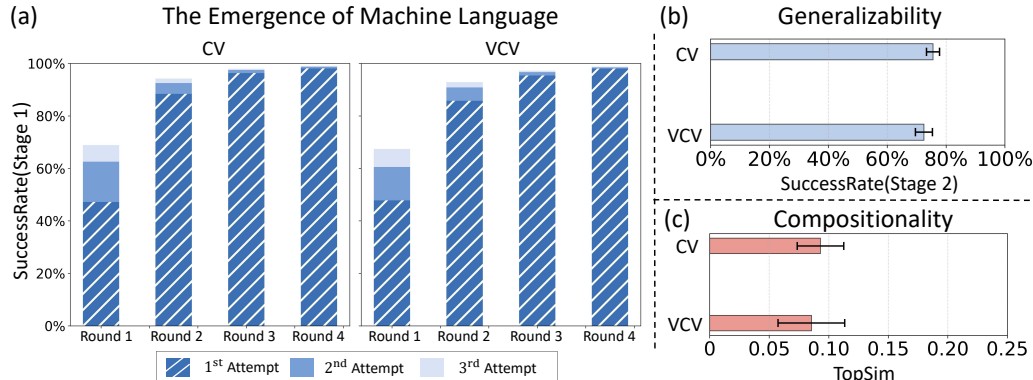

Figure 3: Language emergence across 541 objects under two alphabet settings.(a) Stage 1 (Repeated Communication): Success rates over four rounds, with blue shades indicating success rates at different attempt numbers. (b) Stage 2 (Generalizability): Success rates for previously unseen objects. (c) Compositionality of the emergent language evaluated using topographic similarity. Each experiment was repeated 10 times.

word "chicken" can be represented in CV pairs as "su-su-ke-pe," while in VCV triplets it can be represented as "une-ile-ahu-aji-anu-eze".

**Evaluation Metrics.** To evaluate whether a machine language emerges and to characterize its properties, we consider two metrics:

(i) *Success rate*, the proportion of successful communications in each stage. A communication is successful if the speaker sends a symbolic message describing an object and the listener correctly identifies the target among four distractors. The success rate in Stage 1 indicates whether a machine language has emerged, while the success rate in Stage 2 evaluates whether the emergent language generalizes to previously unseen objects.

(ii) *Topographic similarity (TopSim)*, a common measure of compositionality in emergent machine language, captures the structural correspondence between meanings and symbols (Lazaridou et al., 2018). It is computed as the Spearman correlation between the rankings of pairwise distances in semantic and symbol spaces. We use the *Hamming distance* (Hamming, 1950) for object meanings and the *Levenshtein distance* (Levenshtein, 1965) for the corresponding words. The correlation ranges from $-1$ (perfect negative) to $1$ (perfect positive), with values near $0$ indicating no association.

## 5.2 EMERGENT PHENOMENA OF MACHINE LANGUAGE

**Machine language successfully emerges.** Our most significant finding is that, given 400 objects, the two agents successfully develop a shared language after only four rounds of communication, with at most three attempts per object. Figure 3(a) shows the emergence process in Stage 1 under both alphabet settings. At first, agents succeed in communicating about half of the objects in a single attempt, often requiring two or three attempts to identify the target object. As rounds progress, success rates rise quickly—surpassing 90% by the third round and reaching 100% in the final round. By the end, agents successfully communicate each object in a single attempt, demonstrating the emergence of a machine language independent of the predefined alphabet.

**Emergent machine language generalizes to unseen objects.** With the emergence of machine language, agents can immediately communicate about previously unseen objects. Figure 3(b) shows one-round generalizability performance across 141 unseen objects in Stage 2: 76% success with the CV-pair alphabet and 74% with the VCV-triplet alphabet. These results show that the emergent language generalizes efficiently to novel objects—capturing a defining feature of human language.

**Emergent machine language exhibits compositionality.** The emergent language shows a structural correspondence between meanings and symbols. Figure 3(c) presents topographic similarity, with the highest value being 0.123 under the CV-pair alphabet and 0.134 under the VCV-triplet

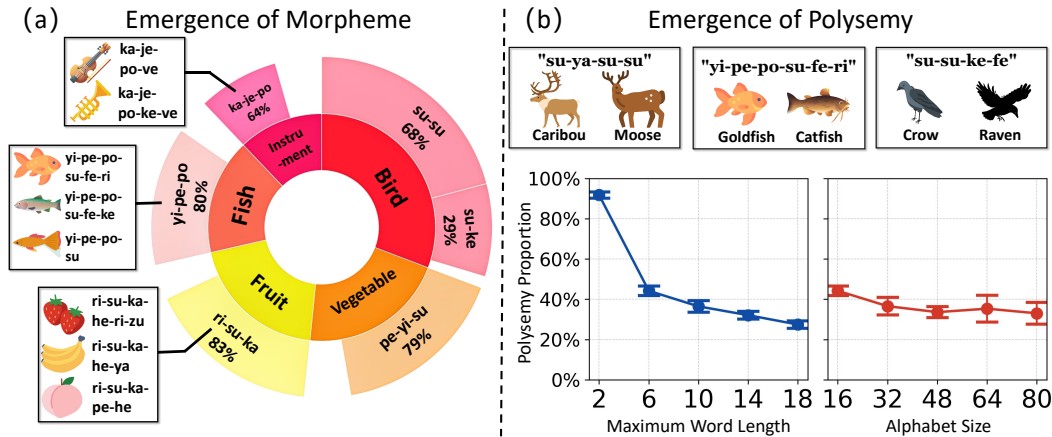

Figure 4: Case studies on the emergence of morphemes and polysemy. (a) Distribution of dominant morphemes across five semantic categories, with examples of symbolic words sharing the same morpheme. (b) Examples of polysemy, where a single word represents multiple related objects. The line plots illustrate how maximum word length (blue) and alphabet size (red) affect the prevalence of polysemy. Results are averaged over five runs, with error bars denoting 95% confidence intervals.

alphabet.[1]. These values indicate a positive correlation between semantic and symbolic distances, quantitatively reflecting the compositional quality of the emergent language.

**Emergence of morphemes.** Morphemes are the smallest units of meaning that combine to form a wide range of words. In natural language, morphemes such as the plural suffix *-s* in "cats" or the prefix *un-* in "unhappy" systematically mark concepts like "more than one" and "not". Figure 4(a) shows the distribution of dominant morphemes across five categories, revealing that certain morphemes cluster strongly within specific semantic domains. For example, the morpheme "yi-pe-po" occurs in 80% of fish-related objects: "yi-pe-po-su-fe-ri" represents goldfish, "yi-pe-po-su-fe-ke" represents trout, and "yi-pe-po-su" represents guppy. Similarly, "ri-su-ka" appears in 83% of fruit-related objects: "ri-su-ka-he-ri-zu" represents strawberry, "ri-su-ka-he-ya" represents banana, and "ri-su-ka-pe-he" represents peach. These patterns highlight systematic symbol–meaning correspondences, suggesting that emergent machine languages can develop morpheme-like structures analogous to those in human languages.

**Emergence of polysemy.** Polysemy refers to the phenomenon where a single word represents multiple related objects. Figure 4(b) illustrates several examples: "su-ya-su-su" denotes both *Caribou* and *Moose*, while "yi-pe-po-su-fe-ri" refers to both *Goldfish* and *Catfish*, and "su-su-ke-fe" applies to *Crow*, *Raven*. These cases show that emergent machine languages can develop polysemy, with the same symbol systematically referring to semantically related objects. We further note that the prevalence of polysemy decreases as alphabet size and word length increase.

**Effect of maximum word length.** We evaluated five maximum word lengths ($\mathcal{L} \in 2, 6, 10, 14, 18$) and found that word length strongly influences the quality of the emergent language. Figure 5(a) shows variation in generalizability success rates and topographic similarity across conditions. With $\mathcal{L} = 2$, the Stage 2 success rate is 61%, indicating that machine language hardly emerges. As maximum word length increases, success rates rise from 61% to 77%, while topographic similarity increases on average from 0.060 to 0.127. These findings suggest that longer words provide greater expressive capacity, thereby supporting the emergence of higher-quality machine languages.

**Effect of alphabet size.** We evaluated five alphabet sizes ($\mathcal{A} \in 16, 32, 48, 64, 80$) and found that alphabet size also affects the quality of the emergent language. Figure 5(b) shows variation in generalizability success rates and topographic similarity (TopSim) across conditions. As alphabet

---

[1]In prior MARL-based studies on machine language emergence, this value typically ranges from 0 to 0.16 for datasets comparable to ours (Lazaridou et al., 2018)

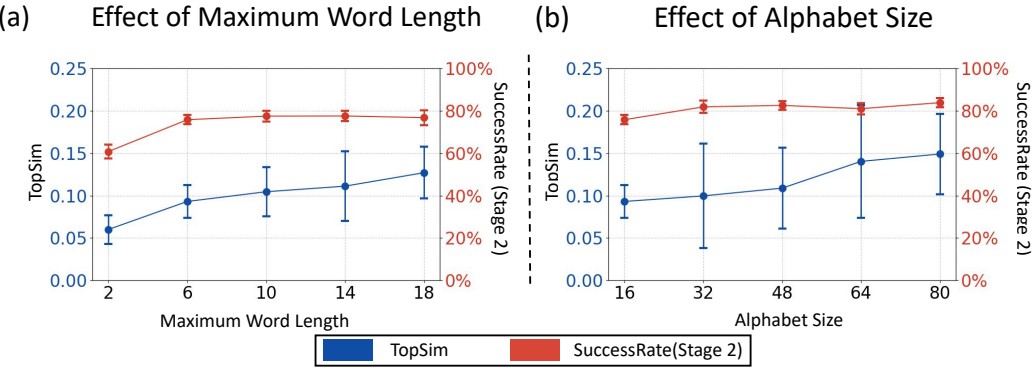

Figure 5: Effects of maximum word length and alphabet size on language emergence. Panel (a) depicts the influence of maximum word length. Panel (b) depicts the Influence of alphabet size. The red line shows the success rate in generalizability (Stage 2), and the blue line shows the topographic similarity (TopSim) of the emergent language. Results are averaged over 10 runs, with error bars denoting 95% confidence intervals.

size increases, success rates rise from $75.8\%$ to $83.8\%$, and topographic similarity improves from $0.093$ to $0.149$. These findings indicate that larger alphabets offer greater compositional flexibility, thereby supporting the emergence of higher-quality machine languages.

Table 1: Effect of object number on language emergence. The table shows the accumulated success rate across attempts in Round 1 of Stage 1, the success rate in Stage 2 and the corresponding topographic similarity (TopSim) of the emergent language. Results are averaged over 10 runs. Bold values indicate the best performance across all metrics.

| Object Number | Accumulated Success Rate (Round 1, Stage 1) | | | Success Rate | TopSim |
| | Attempt1(%) | Attempt2(%) | Attempt3(%) | (Stage 2,%) | |
|---|---|---|---|---|---|
| 100 objects | $(27.7 \pm 6.7)$ | $(40 \pm 7.4)$ | $(47.6 \pm 8.0)$ | $(48.6 \pm 3.4)$ | $0.084 \pm 0.139$ |
| 200 objects | $(37.7 \pm 3.6)$ | $(51.8 \pm 3.9)$ | $(58 \pm 3.6)$ | $(64.6 \pm 4.8)$ | $0.087 \pm 0.028$ |
| **400 objects** | $\mathbf{(47.2 \pm 2.8)}$ | $\mathbf{(62.6 \pm 2.4)}$ | $\mathbf{(68.9 \pm 2.6)}$ | $\mathbf{(75.8 \pm 2.9)}$ | $\mathbf{0.093 \pm 0.026}$ |

**Effect of object number.** We varied the number of objects $(100, 200, 400)$ and found that larger sets promote the emergence of a shared language. Table 5.2 shows that with $100$ objects, the first-round success rate after three attempts in Stage 1 is $47.6\%$, leaving over half of the objects unable to communicate using a shared language. As the number of objects increases, success rates rise (from $47.6\%$ to $68.9\%$ in Stage 1, and from $48.6\%$ to $75.8\%$ in Stage 2), and the mean of topographic similarity increases from $0.084$ to $0.093$. These results show that larger object sets drive agents to have faster rate of language emergence and develop higher-quality machine languages.

## 6 CONCLUSION

The study of language emergence has been a longstanding area of research in AI, particularly within MARL. Yet, it remains an open challenge to enable agents to develop a structured, meaningful, and generalizable machine language with characteristics comparable to natural language. LLMs, given their strong capabilities in natural language (Kumar, 2024; Xu et al., 2024), provide a unique opportunity to investigate whether a machine language—potentially not human-interpretable—can emerge through agent communication. In this paper, we study language emergence through referential games and propose an agent design for LLM-based agents, where the speaker retrieves semantically related words before composing a message and the listener decodes it based on structural proximity. Our experiments show that a machine language emerges with 400 objects and generalizes to 141 previously unseen objects. Analysis further reveals hallmarks of natural language, including compositionality, morphemes, and polysemy. Finally, we examine how factors such as alphabet size, maximum word length, and number of objects affect the quality of emergent language.

## REPRODUCIBILITY STATEMENT

We implemented our proposed communication protocol and experiments in Python. Implementation details are provided in Section 5.1 (Experimental Settings), with additional information in Appendix A to facilitate reproduction. Anonymous URLs to our code are included, enabling the research community to fully access, reproduce, and extend our work on language emergence. The repository also contains detailed instructions for reproducing our results.

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

## A  MODEL PARAMETERS AND PROMPTS

In Section 5.1 of the main paper, we have already outlined the experimental setup. Next, we will present more details of the experimental setup, including model parameters and prompts for LLM. **Model Parameters.** All LLM based modules generate content using the following parameters: $top\_p = 1$, $frequency\_penalty = 0$, $presence\_penalty = 0$, $temperature = 0.1$. Constrained by these parameters, the model will output more deterministic and less random content. **Prompt.** Inspired by CoT (Wei et al., 2022), we require the agent to first analyze the task they are facing in the prompt and then output the final content. Here are the prompts for LLM based operations in our communication protocol.

---

**Semantic-based Retrieval**

---

**TASK：** By determining the semantic features of an object, we can obtain an objective description of that object. Next, we will provide you with a **GIVEN OBJECT'S SEMANTIC FEATURES**. Then you need to select objects from the **ALL OBJECTS' SEMANTIC FEATURES** that you think are semantically similar or closely related to the given object.

**GIVEN OBJECT'S SEMANTIC FEATURES：** JSON

  {'encyclopaedic': ... ,
  'function': ... ,
  'smell': ... ,
  'sound': ... ,
  'tactile': ... ,
  'taxonomic': ... ,
  ... }

**ALL OBJECTS' SEMANTIC FEATURES：** JSON

  Object_i : {'encyclopaedic': ... ,
  'function': ... ,
  'smell': ... ,
  'sound': ... ,
  'tactile': ... ,
  'taxonomic': ... ,
  ... }
  ......

**REQUIREMENTS:** (i) Before completing the task, please conduct a brief analysis; (ii) follow the **EXPECTED FORMAT**.

**EXPECTED FORMAT: JSON**

{
  "analysis":"<Provide your analysis of the current task>",
  "object_list":[<List selected objects >]
}

---

Figure 6: Prompt for Semantic-based Retrieval

---

**Message Generation&Reflection-based Regeneration**

---

**TASK：** You are studying a special artificial language that has <letter num> letters. You must generate exactly one word in this artificial language to describe the current object, using only the **LETTERS** listed. Your task is to generate a word to express **OBJECT FEATURES**, which will be sent to other language learner for him to guess the current **OBJECT FEATURES**. You will be given your own **LEARNED VOCABULARY**, which can help you generate correct words. In addition, this is a multi round communication process, and the **FAILED COMMUNICATION RECORDS** between you and your partner will be provided. It is a dictionary in JSON format, with the key being a word you have used before and the value being your partner's incorrect understanding of the word.

**LETTERS:** <letter list>
**GIVEN OBJECT'S SEMANTIC FEATURES：** JSON
 {'encyclopaedic': ... ,
 'function': ... ,
 'smell': ... ,
 'sound': ... ,
 'tactile': ... ,
 'taxonomic': ... ,
 ... }
**LEARNED VOCABULARY:** JSON
 'Object_i' : {'encyclopaedic': ... ,
 'function': ... ,
 'smell': ... ,
 'sound': ... ,
 'tactile': ... ,
 'taxonomic': ... ,
 ... }
 ......
**FAILED COMMUNICATION RECORDS:** JSON
 'Object_i' : {'encyclopaedic': ... , ...}
 ...
**REQUIREMENTS:** (i) Before completing the task, please conduct a brief analysis; (ii) follow the **EXPECTED FORMAT**.
**EXPECTED FORMAT: JSON**
{
   "analysis":"<Provide your analysis of the current task>",
   "word":<Generated word>
}

---

Figure 7: Prompt for Message Generation

---

**Structural Proximity-based Retrieval**

---

**TASK：** You are a language learner. You are studying a special artificial language that has <letter num> letters. You can see them in the **LETTERS** section. Your task is to select words from the **LEARNED VOCABULARY** that have similarity in word form to the **GIVEN WORD**, so that you believe they have a semantic connection with the **GIVEN WORD**.
**LETTERS:** <letter list>
**GIVEN WORD：** <Word from speaker>
**LEARNED VOCABULARY：** JSON
 Object_i : <Corresponding word>
 ......
**REQUIREMENTS:** (i) Before completing the task, please conduct a brief analysis; (ii) follow the **EXPECTED FORMAT**.
**EXPECTED FORMAT: JSON**
{
   "analysis":"<Provide your analysis of the current task>",
   "object_list":<List selected objects>
}

---

Figure 8: Prompt for Structural Proximity-based Retrieval

---

**Object Selection**

---

**TASK:** You are a language learner. You are studying a special artificial language that has <letter num> letters. You can see them in the **LETTERS** section. Your task is to, upon receiving a **GIVEN WORD** in the current artificial language, select the most likely **SEMANTIC FEATURES** that reflect the meaning of the **GIVEN WORD**, based on **LEARNED VOCABULARY**.
**LETTERS:** <letter list>
**GIVEN WORD:** <Word from speaker>
**LEARNED VOCABULARY:** JSON
  Word_i : {'encyclopaedic': ... ,
  'function': ... ,
  'smell': ... ,
  'sound': ... ,
  'tactile': ... ,
  'taxonomic': ... ,
  ... }
  ......
**SEMANTIC FEATURES**: JSON
  A : {'encyclopaedic': ... ,
  'function': ... ,
  'smell': ... ,
  'sound': ... ,
  'tactile': ... ,
  'taxonomic': ... ,
  ... },
  B : {'encyclopaedic': ... , ...}
  ......
**REQUIREMENTS:** (i) Before completing the task, please conduct a brief analysis; (ii) follow the **EXPECTED FORMAT**.
**EXPECTED FORMAT: JSON**
{
  "analysis":"<Provide your analysis of the current task>",
  "option": <Directly return options from "A","B", etc.>
}

---

Figure 9: Prompt for Object Selection

## B  OBJECT REPRESENTATION

In our setup, objects are represented as a combination of features, and their natural language representation is not provided to the agent. This allows the agent to capture the semantic information of the object without being influenced by the grammatical priors of natural language. McRae et al. (2005) proposes a dataset containing 541 objects that exist in the objective world, where each object is labeled with features by human participants, which reflects human semantic memory of these objects. These features are grouped into ten categories using *brain region taxonomy*, allowing us to use a standardized approach to represent objects. Some examples are shown below.

---

**Some examples of object representations**

---

**"ANT":** "encyclopaedic": ["is strong", "lives in a colony", "lives in a hill", "lives in ground"],
"function": [],
"smell": [],
"sound": [],
"tactile": [],
"taste": [],
"taxonomic": ["an insect"],
"visual_colour": ["is black", "is red"],
"visual_form_and_surface": ["has 6 legs", "has antennae", "is small"],
"visual_motion": ["living behavior: bites", "living behavior: crawls"]

---

Figure 10: Object Representation Examples.1

---

**Some examples of object representations**

---

**TABLE:** encyclopaedic: [found in dining rooms, found in kitchens, has chairs],
function: [used for eating on],
smell: [],
sound: [],
tactile: [],
taste: [],
taxonomic: [furniture],
visual_colour: [],
visual_form_and_surface: [has 4 legs, has legs, is flat, is round, is square, made of wood],
visual_motion: []
**COAT:** encyclopaedic: [worn for winter],
function: [worn for covering, worn for protection, worn for the cold, worn for warmth],
smell: [],
sound: [],
tactile: [is warm],
taste: [],
taxonomic: [clothing],
visual_colour: [different colours ],
visual_form_and_surface: [has a hood, has a zipper, has buttons, has pockets, has sleeves, is long, made of cotton, made of
different materials, made of fur, made of leather, made of wool],
visual_motion: []
**DISH:** encyclopaedic: [found in kitchens, is breakable ],
function: [requires washing, used for eating, used for food],
smell: [],
sound: [],
tactile: [],
taste: [],
taxonomic: [],
visual_colour: [different colours],
visual_form_and_surface: [is round, made of ceramic, made of glass, made of plastic],
visual_motion: []
**BOOK:** encyclopaedic: [found in libraries, found in schools, found on shelves, has authors, non-living behavior: tells stories],
function: [used by reading,used for acquiring/storing knowledge,used for learning],
smell: [],
sound: [],
tactile: [],
taste: [],
taxonomic: [],
visual_colour: [],
visual_form_and_surface: [has a hard cover, has a soft cover, has information, has page numbers, has pages, has pictures, has
words in it, made of paper],
visual_motion: []

---

Figure 11: Object Representation Examples.2

## C   THE USE OF LARGE LANGUAGE MODEL

In this study, all the writing, analysis, and viewpoint elaboration of the main content were independently completed by the author, without the use of any LLM for auxiliary generation or text editing. However, during the literature search and preliminary screening stage, we used LLM tools to conduct exploratory searches on keywords, research context, and some classic literature in related fields to assist in sorting out the research background. All clues obtained through this channel have undergone strict academic verification and have been confirmed by reviewing original literature through formal channels, ensuring the rigor of the research process and the accuracy of citations.

