# OpenReview forum: "Emergence of Machine Language in LLM-based Agent Communication"
_ICLR.cc/2026/Conference — ICLR 2026 Conference Withdrawn Submission_

### Official Review · Reviewer_LQV4 · 2025-10-15

**Soundness:** 3
**Presentation:** 3
**Contribution:** 2
**Rating:** 6
**Confidence:** 4

**Summary:**

## Summary
This paper explores the emergence of artificial (machine) language in communication between large language model (LLM)-based agents. The authors propose a memory-based learning framework that allows agents to develop natural-like but non-human-interpretable communication protocols. The study analyzes the properties of the emergent language and shows that it generalizes to unseen objects and exhibits natural-language-like features such as compositionality.

The topic is original and well motivated, and the paper is clearly written and well structured. The use of pretrained LLMs as agents introduces an interesting angle to the emergent communication (EmComm) field. However, the methodological innovation and analysis depth remain limited, and several conceptual and empirical aspects need clarification or extension.

**Strengths:**

## Strengths
- The paper introduces an interesting and innovative approach for generating natural-like communication that is not easily interpretable by humans.
- The authors demonstrate the ability of the emergent language to generalize to unseen objects.
- The paper presents preliminary evidence for the emergence of machine-language traits that resemble characteristics of natural language.
- The framing of machine-language emergence in LLM-based agents is novel relative to standard symbol-based emergent communication studies.
- The writing and structure are clear and accessible.

**Weaknesses:**

## Weaknesses
- The key distinction between this work and the extensive literature on emergent communication (EmComm) is not clearly articulated.
  In prior works, learning is performed end-to-end through reinforcement learning (RL) or straight-through gradient estimation, whereas this paper relies on a memory-based learning mechanism that updates after successful interactions.
  The paper should explicitly clarify and discuss these differences, ideally through direct comparison with past methods.

- The dependence on pretrained LLMs, which already encode extensive natural language knowledge, is insufficiently analyzed.
  The implications of relying on models trained on vast natural-language corpora, particularly when object descriptions are expressed in natural language, should be more deeply discussed.
  The paper does not convincingly address whether emergent communication is genuinely novel or merely a reorganization of existing linguistic priors.

- The motivation for creating a machine language, as well as its potential benefits and risks, is not sufficiently explored.
  The discussion should contrast the advantages and limitations of the emergent language relative to natural language.

- The evaluation of natural-language-like properties (e.g., compositionality, vocabulary size, word length) relies solely on topographic similarity (TopSim), which is known to have severe limitations.
  Adding additional compositionality metrics such as AMI (Mu & Goodman, 2021), CBM (Carmeli et al., 2024), and Context Independence (CI) (Bogin et al., 2018) would significantly strengthen the credibility of the analysis.

---

### Related Work
A substantial body of EmComm research has examined agents’ ability to generate communication protocols from scratch, many without using reinforcement learning (e.g., Choi et al., 2018; Carmeli et al., 2025; Tucker et al., 2022).
These studies typically begin with **random symbol vocabularies** and learn mappings through differentiable or obverter-style updates.
It is unclear how the proposed **LLM-based communication** framework fundamentally differs from these settings.
The authors should explicitly cite and discuss these prior works, clarifying how their approach contributes beyond them.

---

### References

**Referential Games (beyond RL):**
- Choi, E., Lazaridou, A., & De Freitas, N. (2018). *Compositional obverter communication learning from raw visual input.* ICLR.
- Carmeli, B., Meir, R., & Belinkov, Y. (2025). *Composition through decomposition in emergent communication (CtD).* ICLR.
- Tucker, M., Levy, R., Shah, J., & Zaslavsky, N. (2022). *Trading off utility, informativeness, and complexity in emergent communication (VQ-ViB).* NeurIPS 35, 22214–22228.

**Compositionality Metrics:**
- Mu, J., & Goodman, N. (2021). *Emergent communication of generalizations.* NeurIPS 34, 17994–18007.
- Carmeli, B., Belinkov, Y., & Meir, R. (2024). *Concept-best-matching: Evaluating compositionality in emergent communication (CBM).* arXiv:2403.14705.
- Bogin, B., Geva, M., & Berant, J. (2018). *Emergence of communication in an interactive world with consistent speakers (Context Independence).* arXiv:1809.00549.

---

**Questions:**

## Specific Comments and Questions

- **Lines 148–158:** The paragraph describing object properties (e.g., smell, tactile, taste) seems disconnected from the actual implementation.

- **Line 151:** Reference required.

- **Line 221:** Please clarify the phrase *“both agents update their memories”*.  If communication is successful, why is memory updated, and what information is stored?

- **Section 4.2:** How is object similarity computed? Is it derived from the LLM’s embedding space, and if so, does this rely on prior natural-language knowledge?

- **Lines 234–236:** The claim that using *semantic features rather than lexical labels* reduces LLMs’ reliance on linguistic priors is questionable. Semantic features themselves are grounded in natural language-based representations.

- **Lines 273–274:** How many distractors are sampled in each communication round? Are all stored objects considered, or only a subset? If a subset then how are they chosen?

- **Lines 276–284:** Please clarify the reward mechanism.  The paper indicates that learning occurs only after successful communication. Would the system benefit from incorporating negative feedback on failed interactions?

- **Line 292:** Suggested rewording:
  > “We answer the above-mentioned questions in Section 5.2.”

- **Line 308:** The reference to *“the brain region taxonomy”* in McRae et al. (2005) is misleading.
  McRae et al. proposed a **semantic feature taxonomy**, not a brain-region taxonomy.
  Please revise for accuracy.

- **Lines 318–323:** Regarding alphabet design — do the two vocabulary types (CV and VCV) meaningfully affect learning? Did the authors test simpler alternatives such as integer tokens, as in prior work?

- **Lines 355–359:** Many established EmComm metrics exist beyond TopSim, which is widely regarded as unreliable.
  Please consider including **AMI**, **CBM**, or **Context Independence (CI)** for a more robust compositionality analysis.

- **Lines 363–374:** The generalization findings are more compelling than the communication results on 400 objects.
  The baseline accuracy (likely 0.25 for four candidates) should be reported for context.

- **Lines 424–425:** The statement that *“longer words provide greater expressive capacity”* is self-evident but does not necessarily imply *higher-quality* languages.
  Greater channel capacity may lead to communicating low-level features rather than improved compositionality.

- **Lines 451–453:** The conclusion that larger vocabularies yield better languages is not supported by either the presented data or prior literature.
  Typically, larger vocabularies improve task accuracy but reduce compositionality; this trade-off should be discussed.

---

### Official Review · Reviewer_gPaF · 2025-10-21

**Soundness:** 2
**Presentation:** 2
**Contribution:** 1
**Rating:** 2
**Confidence:** 5

**Summary:**

This paper demonstrates that pretrained, frozen LLMs can construct human uninterpretable machine language while playing a Lewis game in an ICL setting. This language exhibits several traits present in human languages, such as weak compositionality, morphology, and polysemy. Longer words, larger vocabulary size, and object sets promote higher compositionality.

**Strengths:**

This paper explores an interesting topic, which is the emergence of human-uninterpretable machine language from interactions. Overall, the methodology is clear and the experiments seem reproducible (prompts and code are provided-- I did not test the code). The representation of objects is scaled up compared to prior work (Kouwenhoven et al., 2025). Still, this paper is very similar to Kouwenhoven et al., see Weaknesses.

**Weaknesses:**

The current submission is very similar to Kouwenhoven 2025. Further, there are several claims regarding machine language emergence which are encouraged in the experimental design (hence not emergence).

### Injecting priors into experimental design, then claiming emergence of those priors
1. Asking the listener to retrieve words based on structural proximity **encodes the prior** that similar words should have similar meanings.
2. This also encourages the emergence of morphemes
3. line 140-141 "In contrast to our work, which explores language emergence from scratch, these studies assume a pre-defined vocabulary for each agent" This is also true for this paper, where the alphabet $\mathcal A$ is pre-defined.

### Too similar to Kouwenhoven 2025
My primary reason to reject is for **lack of novelty with respect to Kouwenhoven2025a** (not cited by the authors) and Kouwenhoven2025b (cited in l136 but not meaningfully engaged with). I do not think it is realistic to reshape this paper to significantly depart from theirs in the current review timeframe. With some tweaks ACL can be a great choice of venue

Kouwenhoven2025a (to the best of my knowledge) were the first to show that LLMs develop "machine language" in an ICL setting. The current submission is extremely similar to that paper. Here are several of the similarities:

1. **Attribute-based representations of the objects**, e.g., color, shape, number. The present paper greatly scales up the representation using the brain taxonomy representation, which includes more attributes and values. Still, this experimental setting does not significantly depart from the attribute-value setting used since at least 5 years now [Chaabouni2021], and we still fall short from implementing machine language for unseen objects in-the-wild if we rely only on the brain taxonomy for features.

2. **ICL setting** rather than MARL

3. Using **CV-style syllables** as the vocabulary (the present paper also uses CVC, but I'm not sure that this adds anything to the overall message)

4. Very **similar prompts** to Kouwenhoven2025a.

5. Very **similar conclusions** including the emergence of morphology (Kouwenhoven2025a Section 5.4) and homonymy (Kouwenhoven2025b). The analysis done in Kouwenhoven2025a and Kouwenhoven2025b is more sophisticated, e.g., they quantify the amount of homonymy and show its evolution over time, in addition to the qualitative analysis. I would recommend doing the same.

6. Kouwenhoven2025a does not explicitly instruct agents to use structural similarity for object retrieval, instead showing it to be an emergent property. I find this much more compelling than explicitly instructing the LLMs to do so. I would recommend adding an experiment ablating the explicit request to use structural similarity.

### Other:

- I'm missing a discussion of in-context learning, which is what agents are based on.
- Only gpt-4-mini was used for experiments. Given that the present paper is too similar to Kouwenhoven2025, I would experiment on different LLMs or populations of LLMs for a future iteration.
- A topsim of between 0 and 0.15 (as reported in the paper) is not very high-- I would not claim the language is compositional based on these values. Indeed, Chaabouni et al., 2020 claim that emergent languages are **not compositional** using a similar range of values (~0.11).

### Missing related work:
[Kouwenhoven2025a] Searching for Structure: Investigating Emergent Communication with Large Language Models, Kouwenhoven et al., [COLING 2025](https://openreview.net/forum?id=kst43TfV9b)

Emergent communication at scale, Chaabouni et al., ICLR 2022.

### Bibliography
[Kouwenhoven2025b] Kouwenhoven et al., IJCAI 2025

[Chaabouni2021] Compositionality and generalization in emergent languages, Chaabouni et al., 2021.

**Questions:**

Here are some minor weaknesses (did not contribute to my score).

l031: remove Peters et al., 2025 and Lazaridou and Baroni, 2020-- cite something more general like Hockett, 1960

l015 stylish paradigm -> popular paradigm

l071 Does "majority of 400 objects" mean at least 200 objects? To me, this seems rather permissive.

l201 Generalizability does not have anything to do per-se with "hierarchical description of concepts and relations"-- I would remove mention of hierarchy here.

l321 "each letter being a syllable" -> letter should be "element". letters are "a", "b", "c", ... and syllables are composed of letters.

l359 Levenshtein distance between strings or symbol sequences?

---

### Official Review · Reviewer_f3Bi · 2025-10-26

**Soundness:** 1
**Presentation:** 3
**Contribution:** 1
**Rating:** 2
**Confidence:** 4

**Summary:**

This paper explores the ability of two LLM agents to learn effective strategies in a Lewis reference game. The setup is as follows:
- There is a set of objects known to both agents. Each object has a list of features in natural language. E.g. "Ant: Taxonomy: 'an insect', Colour: ["black", "red"],...".
- There is a fixed alphabet (e.g. consonant-vowel such as "va", "ca", "bi") and maximal word length $L$ known to both agents.
- Two agents are initialized with an empty memory. The memory maps objects to words, which are strings of length at most $L$ over the alphabet.
- The experiment is carried out as follows. For each round in a set number of rounds, for each object $O$ in the list of objects:
 - One agent is chosen as Speaker and the other as Listener, at random. The Speaker generates a word $w$ based on its memory.The other agent (the "listener"), receives $w$ and a set of objects consisting of $O$ and distractors. The listener then selectsan object $O'$ based on its own memory. If $O = O'$, continue to the next object. Else, repeat this (generating $w$ and $O'$) at most two more times. If $O \neq O'$ in all three of the attempts, the object is skipped.

The word generation $w$ as well as the object selection $O'$ are done via prompting. The authors explore several design choices for the game, e.g. the effect of consonant-vowel alphabet vs. vowel-consonant-vowel alphabet. The authors find that agents are able to achieve agreement on all objects within four rounds, i.e., four passes through all objects. They also find that agents are able to generalize successful communication to unseen objects based on the new objects' features. Furthermore, objects with similar features (small Hamming distance of the feature vector) have similar words (small edit distance).

**Strengths:**

- Originality: To my knowledge, this is the first paper to bootstrap pretrained LLMs to play Lewis reference games, and evaluate the emergent communication protocols.
- Clarity: The bootstrapping process generally well-presented and the high-level is easy to understand. I believe is approachable even to readers with no experience in emergent communication. The claimed results are clearly stated, and the evaluation methods are clear.
- Quality: Basing the work on Lewis games, which are extremely well-studied, is a good starting point as it opens the door to relating the findings to the many related studies.

**Weaknesses:**

- The paper asks whether two LLM agents can be bootstrapped (prompted in a particular scheme) to converge on a winning strategy in Lewis reference games. The answer is yes, and it takes about four passes through the objects. There is, however, a significant gap between this simple setup and the central claim of the paper: the “Emergence of Machine Language.” An long-standing debate in and around the emergent communication community concerns the point at which a learned protocol in a simplified setup can genuinely be called a language. Per Hockett, language should display displacement and true productivity; Chomsky’s Faculty of Language requires recursive compositionality (infinite meanings generated from a finite base set). The standard Lewis reference game setup cannot generate protocols (communication systems) that meet any of these criteria, at least not without a very complicated evaluation suite. I would at least expect to see a serious engagement with these foundational questions if the authors wish to claim such a result.
- Lewis reference games were a suitable choice for stylized experiments at the nascent stage of emergent communication because it was highly non-trivial to get deep reinforcement learning to learn an effective policy from scratch. This paper sidesteps that significant difficulty by initializing the agents with GPT-4.1-mini. This pretrained model is already imbued with extensive linguistic structure and pragmatic competence. From such a starting point, successfully coordinating in a Lewis game is largely expected. This is in contrast to when randomly initialized deep RL agents learn an effective protocol (let alone one that exhibits signs of compositionality). But when agents that already exhibit understanding of human grammar and semantics do so, it is comparable to observing that humans (in fact, polyglots) can coordinate through language in a constrained setting. In other words, I view the claimed findings in this paper (emergence, compositionality, morphemes) as the reuse of existing linguistic priors rather than the emergence of language itself.

**Questions:**

- In emergent communication where both agents are deep NNs, the update after each unsuccessful round is commonly by backpropogation. How is the update carried out here?
- Why did you choose the term Machine Language over Emergent Communication?

---

### Official Review · Reviewer_9z2x · 2025-10-27

**Soundness:** 1
**Presentation:** 3
**Contribution:** 2
**Rating:** 4
**Confidence:** 4

**Summary:**

# Problem:

Can Large Language Models (LLMs) make artificial languages emerge?

# Contributions:

In the context of referential games, this paper proposes (i) LLM-based speaker and listener agent designs and (ii) provides experimental results on the Object dataset from [McRae2005].

**Strengths:**

## Quality:

SQ1: I appreciate the ablation study on the capacity of the communication channel (Figure 5 and related text), and the dataset size (Table 1 and related text).

## Clarity:

SC1: Overall, the paper is well-written and easy to read.

## Originality:

SO1: I appreciate the polysemy and morpheme studies, as they strike me as novel and valuable considerations. The quality of Figure 4.a is also high.

## Significance:

SS1: I think this kind of inquiry around emergent communication but from systems that have developed some natural language fluency already -as opposed to from scratch- is a very interesting direction that might have a strong impact on the subfield of Emergent Communication.

**Weaknesses:**

## Quality:

WQ1: Ambiguous usage of ‘generalization’ in the claims: is it in-distribution or out-of-distribution? is the train-test split actually able to measure it (internal validity?)

I would suggest the authors to start with [1, 2, 3].

I appreciate that ‘generalizability’ is defined around ln200, but it remains ambiguous as it relies on non-clearly-defined words like ‘have learned to describe’ (for the speaker) and ‘understand’ (for the listener). It might help to rely on a metric, for instance zero-shot compositional tests, as used in [Chaabouni2021, 6], which requires specifically-constructed train-test splits around compositions of attributes. From my understanding of Stage 1 vs Stage 2 data split (ln312-313), it consists of a random split, therefore there is no zero-shot compositionality tests being performed in the current experiment.

WQ2: “Machine language is considered to emerge if the two agents achieved successful communication on the majority of 400 objects…” (ln70) : Missing discussion related to [4], which showed that accuracy in an emergent communication game is not necessarily an indication of successful communication; the claim in ln70 thus requires clarification. It might be important to introduce to the current study some positive signaling and positive listening metrics.

WQ3: Despite citation of [Chaabouni2021] the paper does not discuss the choice of compositionality metric, and especially does not explain why only measure topographic similarity while [Chaabouni2021] has shown it to be limited compared to their posdis/bosdis proposal (which is refined in [6]). Moreover, why not considering the recent metric proposed by [8] as well?

WQ4: I think it would be interesting to consider performance depending on the number of parameters of the used LLM. It would also increase the external validity of the experiments if they were performed with both closed and open-source/weights LLMs, as opposed to only using closed gpt-4.1-mini.

## Clarity:

WC1: Figure 3, 4, and 5 lack details about the statistics being reported ( standard error of the mean?).

WC2: As [3] showed that reporting topographic similarity measures on whole dataset vs train set vs test set yield different measures, I think it is important that the current paper clarifies what is the current measure computed on.

WC3: It is unclear to me what are the semantic features $f_o$ that defines a generic object $o$ (ln189).

## Originality:

WO1 : Missing discussion with [3] regarding the results presented in Figure 5. Indeed, [3] found that (i) increasing the maximum sentence/word length is beneficial to further both compositionality and generalisation abilities, but (ii) increasing the vocabulary size is found detrimental.

WO2: Missing citation to [7] around ln215 ( 4.1 Memory).

## Significance:

WS1: Claim 3 (ln92) is made in a vacuum: language emergence is efficient and robust in comparison to what? I would advise the authors to consider adding some common baselines (e.g. [3] or [5]), or some ablation study showing that a specific design choice yields greater efficiency and robustness compared to another design choice.

WS2 : The same critic goes for the second part of the claim regarding generalizability and compositionality: e.g. what is the threshold above which the measured Topographic Similarity can indicate compositionality? I appreciate the footnote information for paragraph starting in ln376, but it makes for a rather weak evidence at best. It would be better to try to measure the compositionality on the relevant dataset (with the same train-test splits) with a common approach, for comparison, maybe?

**Questions:**

Please see Weaknesses section.


# References:

[1] : Lake, Brenden, and Marco Baroni. "Generalization without systematicity: On the compositional skills of sequence-to-sequence recurrent networks." *International conference on machine learning*. PMLR, 2018.

[2] : D. Bahdanau, S. Murty, M. Noukhovitch, T. H. Nguyen, H. de Vries, and A. Courville. “Systematic Generalization: What Is Required and Can It Be Learned?” International Conference on Learning Representations, nov 2019.

[3] : Denamganaï, Kevin, and James Alfred Walker. "On (emergent) systematic generalisation and compositionality in visual referential games with straight-through gumbel-softmax estimator." *arXiv preprint arXiv:2012.10776* (2020). (EmeCom Workshop @ NeurIPS2020)

[4] : Lowe, Ryan, et al. "On the Pitfalls of Measuring Emergent Communication." *Proceedings of the 18th International Conference on Autonomous Agents and MultiAgent Systems*. 2019.

[5] : Auersperger, Michal, and Pavel Pecina. "Defending Compositionality in Emergent Languages." *NAACL 2022* (2022): 285.

[6] : Denamganaï, Kevin, Sondess Missaoui, and James Alfred Walker. "Visual referential games further the emergence of disentangled representations." *arXiv preprint arXiv:2304.14511* (2023).

[7] : Satwik Kottur, José Moura, Stefan Lee, and Dhruv Batra. 2017. [Natural Language Does Not Emerge ‘Naturally’ in Multi-Agent Dialog](https://aclanthology.org/D17-1321/). In *Proceedings of the 2017 Conference on Empirical Methods in Natural Language Processing*, pages 2962–2967, Copenhagen, Denmark. Association for Computational Linguistics.

[8] : Boaz Carmeli, Yonatan Belinkov, and Ron Meir. 2024. [Concept-Best-Matching: Evaluating Compositionality In Emergent Communication](https://aclanthology.org/2024.findings-acl.189/). In *Findings of the Association for Computational Linguistics: ACL 2024*, pages 3186–3194, Bangkok, Thailand. Association for Computational Linguistics.

---

### Author Response · Authors · 2025-11-23

We sincerely thank all the reviewers for their time and efforts reviewing our paper. We found most of the comments to be helpful and constructive, and we plan to substantially revise our paper according to the comments. Thus, we have decided to withdraw it for now and will consider submitting a significantly improved version to another venue.
That said, we would like to clarify a misunderstanding regarding the relationship between our work and that of Kouwenhoven et al.
First of all, our work addresses a **different problem** from that of Kouwenhoven et al. We investigate whether language—specifically, the mapping between words and objects—can **emerge from scratch**. In contrast, their study assumes such a language is already present from the outset; that is, **given a pre-existing language**, they examine how it develops desirable or interesting properties over multiple stages and through inter-agent interactions. Therefore, our work and theirs answer distinct reserach questions, and our contributions lie in a different direction.
Here are some evidence.

In Section 4.1 (Memory), we write:

> “Initially, both agents’ memories are **empty**. ..
>

In Section 4.2 we write:

> “It is non-trivial to communicate using a novel pre-defined alphabet **without any prior linguistic grounding, as agents must autonomously develop a machine language from scratch**. Consequently, communication failures are common, particularly at the beginning when only few object–word pairs are available for reference…”
>

These show that our “machine language” is truly emergent: it starts from no words, no shared conventions, and empty memories, and arises solely through agent interactions.

By contrast, Kouwenhoven et al. give the agents an pre-defined initial vocabulary via the prompt; the object–word mappings are pre-specified, not negotiated by the agents. They write:

> “Initial signals for these stimuli were generated before each experiment.”
>

and

> “the agents observe the training vocabulary with the current stimulus in the guessing block.”
>

In the appendix, the authors also list the prompts, including the full initial vocabulary:

```
{'shape ':2,'colour ':'orange ','amount ':1,'word ':'giniwite '}
{'shape ':3,'colour ':'green ','amount ':1,'word ':'ginisu '}
{'shape ':1,'colour ':'orange ','amount ':2,'word ':'pinisugi '}
{'shape ':3,'colour ':'green ','amount ':3,'word ':'sutepi '}
{'shape ':2,'colour ':'orange ','amount ':2,'word ':'winisu '}
…
```

Thus, unlike our work, Kouwenhoven et al. presuppose an **pre-defined vocabulary** that is made available to **both** agents from the outset.

Therefore, while our work concerns the very emergence of language, theirs concerns the subsequent evolution of an already established one.

Moreover, our experimental settings also differ substantially from those of Kouwenhoven et al. In their experiments, the dataset consists of 27 objects, each defined along 3 feature dimensions (color, shape, and amount). In contrast, our work uses 541 objects, each represented with 10 semantic feature categories grounded in a well-established, comprehensive semantic feature taxonomy.
Within this much richer and more complex semantic space, and under a substantially larger object set, whether a machine-invented language will emerge is far from trivial. The increased dimensionality, diversity, and granularity of our setting fundamentally change the difficulty of the task: the search space for possible word–object mappings grows combinatorially, the inductive biases required for successful communication become more demanding, and the dynamics of emergence are less constrained by feature structures.
Consequently, the conclusions drawn from our experiments speak to the robustness and generality of emergent language phenomena in a way that settings with limited objects and low-dimensional features cannot fully capture.


Overall, our work focuses on whether a machine language, absent from the training data and not directly human-interpretable, can **autonomously emerge** from LLM-agent interactions. We show that, under our framework, agents are able to develop a language over a large-scale set of 400 objects within only four interaction rounds, and that this emergent language further generalizes to 141 unseen objects.
To our knowledge, this is the first study to investigate whether a machine language of this kind can autonomously emerge from interactions among LLM agents, and the first to demonstrate that such a language—covering hundreds of objects and exhibiting desirable properties—can arise so quickly, within only a handful of interaction rounds.
# Reference:

[Kouwenhoven2025a] Kouwenhoven et al. Searching for structure: Investigating emergent communication with large language models.

---

### Note · Authors · 2025-12-17

I have read and agree with the venue's withdrawal policy on behalf of myself and my co-authors.